

# Competition-function tradeoffs in ectomycorrhizal fungi

Holly V. Moeller[1,2] and Kabir G. Peay[3]

[1] Biology Department, Woods Hole Oceanographic Institution, Woods Hole, MA, USA
[2] Ecology, Evolution & Marine Biology, University of California, Santa Barbara, Santa Barbara, CA, USA
[3] Department of Biology, Stanford University, Stanford, CA, USA

## ABSTRACT

**Background.** The extent to which ectomycorrhizal fungi mediate primary production, carbon storage, and nutrient remineralization in terrestrial ecosystems depends upon fungal community composition. However, the factors that govern community composition at the root system scale are not well understood. Here, we explore a potential tradeoff between ectomycorrhizal fungal competitive ability and enzymatic function.
**Methods.** We grew *Pinus muricata* (Bishop Pine) seedlings in association with ectomycorrhizal fungi from three different genera in a fully factorial experimental design. We measured seedling growth responses, ectomycorrhizal abundance, and the root tip activity of five different extracellular enzymes involved in the mobilization of carbon and phosphorus.
**Results.** We found an inverse relationship between competitiveness, quantified based on relative colonization levels, and enzymatic activity. Specifically, *Thelephora terrestris*, the dominant fungus, had the lowest enzyme activity levels, while *Suillus pungens*, the least dominant fungus, had the highest.
**Discussion.** Our results identify a tradeoff between competition and function in ectomycorrhizal fungi, perhaps mediated by the competing energetic demands associated with competitive interactions and enzymatic production. These data suggest that mechanisms such as active partner maintenance by host trees may be important to maintaining "high-quality" ectomycorrhizal fungal partners in natural systems.

## INTRODUCTION

Ectomycorrhizal fungi (EMF), the belowground mutualistic partners of most of the world's temperate tree species, are key regulators of primary production and nutrient remineralization in terrestrial ecosystems. EMF mediate the transfer of water and nutrients from the soil to their host trees (*Brownlee et al.*, *1983*; *Leake et al.*, *2004*) and serve as a pathway for photosynthetically fixed carbon into the soil community (*Talbot, Allison & Treseder*, *2008*). These fungi also affect the extent of carbon storage (*Averill, Turner & Finzi*, *2014*; *Clemmensen et al.*, *2015*) and the rate of nutrient cycling (*Leake et al.*, *2004*; *Koide, Fernandez & Malcolm*, *2014*) belowground.

The species composition of EMF communities affects these functions. For example, some taxa utilize distinct foraging strategies, such as the formation of rhizomorphs which

Corresponding author
Holly V. Moeller,
hollyvm@alumni.stanford.edu

allow the transport of water and nutrients over long distances (*Brownlee et al.*, *1983*; *Agerer*, *2001*; *Agerer*, *2006*), accumulate relatively high amounts of mycelial biomass belowground (*Hobbie*, *2006*), and are linked with efficient nitrogen mobilization and low carbon sequestration (*Clemmensen et al.*, *2015*). EMF that form melanized hyphae may be more tolerant of water stress (*Fernandez & Koide*, *2013*) and resistant to decomposition (*Fernandez & Koide*, *2014*; *Koide, Fernandez & Malcolm*, *2014*) than other taxa. EMF also differ in the extent to which they produce extracellular enzymes that break down soil organic matter and release nitrogen and phosphorus (*Courty et al.*, *2006*; *Courty, Franc & Garbaye*, *2010*; *Jones et al.*, *2010*; *Tedersoo et al.*, *2012*). Because fungal traits are linked through their function to ecosystem processes (*Koide, Fernandez & Malcolm*, *2014*; *Treseder & Lennon*, *2015*), understanding the factors controlling ectomycorrhizal community composition on host tree root systems is of major importance.

While both abiotic (e.g., edaphic environment (*Moeller, Peay & Fukami*, *2014*)) and biotic (e.g., plant community context (*Bogar & Kennedy*, *2013*; *Moeller et al.*, *2015*)) conditions may play a primary role in filtering members of the ectomycorrhizal community, recent work has highlighted the importance of fungal species interactions to ectomycorrhizal community assembly (*Kennedy*, *2010*). Competitive interactions, in particular, may locally structure fungal communities within a host plant's root system, leading to competitive exclusion (*Villeneuve, Le Tacon & Bouchard*, *1991*; *Kennedy & Bruns*, *2005*; *Kennedy, Peay & Bruns*, *2009*) or spatial segregation (*Taylor & Bruns*, *1999*; *Pickles et al.*, *2012*). Over longer time scales, such competitive interactions may contribute to the successional patterns observed within ectomycorrhizal communities (*Mason et al.*, *1983*; *Visser*, *1995*; *Nara et al.*, *2003*), likely in part through a tradeoff between colonization and competitive abilities (*Lilleskov & Bruns*, *2003*; *Kennedy et al.*, *2011*).

Some functional traits such as foraging type (*Peay, Kennedy & Bruns*, *2011*; *Clemmensen et al.*, *2015*) and propagule persistence (*Baar et al.*, *1999*; *Taylor & Bruns*, *1999*) are associated with successional stage; however, it is not clear whether these differences in functionality result in differing competitive abilities. Here, we tested for a tradeoff between competitiveness and nutrient acquisition ability (measured as enzymatic activity) across three different ectomycorrhizal fungal genera. By holding host tree age, inoculum potential, and environmental conditions constant, we experimentally tested three hypotheses.

First, we hypothesized that a dominance hierarchy exists among the three fungal taxa used in our study, *Rhizopogon occidentalis*, *Suillus pungens*, and *Thelephora terrestris*. Based on prior greenhouse experimental work, we expected *R. occidentalis* to be competitively dominant to *S. pungens* (*Kennedy et al.*, *2007*; *Kennedy et al.*, *2011*), and we expected *T. terrestris*, an aggressive seedling colonizer (*Mason et al.*, *1983*; *Velmala et al.*, *2013*), to be competitively dominant to both these species. Second, we expected this dominance hierarchy to be inversely related to fungal enzymatic function. We based this hypothesis on the rationale that fungi experience an energetic tradeoff: they can either produce metabolically costly extracellular enzymes, or they can invest in chemical defenses against their competitors. Third, we hypothesized that this tradeoff would impact host tree seedling growth: seedlings would accrue the highest biomass when associating with the least dominant, most highly enzymatically functional fungi.

## MATERIALS AND METHODS

### Experimental design

We worked with *Pinus muricata* (Bishop Pine) and three ectomycorrhizal fungi (*Rhizopogon occidentalis*, *Suillus pungens*, and *Thelephora terrestris*) known to associate with this tree in its native range (*Peay et al.*, *2007*). We selected these fungi because they are among the most common and abundant in early successional pine forests in Point Reyes National Seashore (PRNS), where we worked (*Peay et al.*, *2007*). We tested the competitive abilities, enzyme expression levels, and effects on seedling growth of these fungi grown in isolation and in competition.

*P. muricata* seeds were obtained from PRNS. Prior to the start of the experiment, seeds were surface sterilized and germinated in autoclave-sterilized perlite. Within 7 days, germinated seedlings were transplanted into conetainers filled with a 50:50 mix of autoclaved sand and soil from PRNS. Spores from each of the three EMF were obtained from sporocarps collected at PRNS. Sporocarps were incubated spore-side down overnight on foil at room temperature. Spores were collected by washing the foil with sterile (distilled, autoclaved) water and refrigerated at 4 °C for three weeks until inoculation. Prior to inoculation, hemocytometer counts followed by serial dilutions in sterile water were used to obtain a concentration of 1,000 spores per mL for each of the three fungal species. We did not conduct spore viability stain assays; however, spore storage time was short (*Bruns et al.*, *2009*), and handling was consistent with other studies using spore inoculum from these species (*Kennedy & Bruns*, *2005*; *Kennedy & Peay*, *2007*; *Kennedy, Peay & Bruns*, *2009*; *Peay et al.*, *2012*), so we expected a high proportion of viable spores in the inoculation slurries.

We inoculated the *P. muricata* seedlings with zero, one, two, or three fungal species in all possible combinations. We randomly assigned five seedlings to each treatment group (for a total of 5 seedlings × 7 treatments + 1 control = 40 seedlings). Each seedling received a total of 3-mL of inoculum. For control (non-mycorrhizal) seedlings, this consisted of 3-mL of sterile water. Seedlings inoculated with EMF received 1-mL of inoculum per fungal species (so that seedlings in the single-fungus treatments received 1-mL of spore inoculum and 2-mL of distilled water). Thus, seedlings in the three-species treatment received a total of 3,000 spores (1,000 per species).

Seedlings were maintained in a greenhouse at Stanford University for five months between inoculation and harvest. This experimental duration was chosen because it approximates the length of the main growth and fruiting season of EMF in PRNS, and because previous greenhouse studies of *P. muricata* seedlings and their EMF have been of similar duration (*Kennedy & Bruns*, *2005*; *Kennedy & Peay*, *2007*; *Peay, Garbelotto & Bruns*, *2009*; *Kennedy, Peay & Bruns*, *2009*). At harvest, each seedling's root system was separated from the shoot at the root collar. The root system was washed clear of adhering soil using tap water. Roots were then cut into 3-cm segments, homogenized, and a subset was examined under a dissecting microscope. This entire subset was scored for mycorrhization using the grid-line intersection method: root segments were randomly arranged over a 1-cm grid, and every grid crossing was scored as mycorrhizal or non-mycorrhizal based on the presence or absence of fungal hyphae (*Giovannetti & Mosse*, *1980*). Note that, because

not all of the root system is comprised of fine root tips, total mycorrhization levels are always <100%. Following examination, eight mycorrhizal root tips from each seedling were randomly selected for fungal identification, and ten mycorrhizal tips were selected for enzyme assays. Enzyme assay tips were also collected from control seedlings to establish non-mycorrhizal baseline activity levels. Following root system processing, seedling root systems and shoots were dried at 65 °C for 48 h and then weighed to determine biomass.

## Fungal identification and enzyme assays

We used Sanger sequencing to assign mycorrhizal root tips to species. To extract DNA, tips were heated in 10 µL Extraction Solution (SKU E7526; Sigma-Aldrich Co. LLC, St. Louis, MO, USA) for 10 min at 65 °C, then 10 min at 95 °C, before addition of 10 µL of Neutralization Solution B (SKU N3910; Sigma-Aldrich Co. LLC). The internal transcribed spacer (ITS) region of the nuclear ribosomal RNA genes of each root tip was amplified using the ITS-1F (*Gardes & Bruns*, *1993*) and ITS-4 primers (*White et al.*, *1990*) and sequenced by Beckman Coulter Genomics (Danvers, MA, USA). The resultant sequences were assigned to one of the three species using the Basic Local Alignment Search Tool (BLAST, http://blast.ncbi.nlm.nih.gov).

We used fluorimetric assays to quantify the activity levels of five different enzymes: $\alpha$-glucosidase (which hydrolyzes starch and glycogen), $\beta$-glucosidase (which hydrolyzes cellobiose), N-acetyl-glucosaminidase (which breaks down chitin), $\beta$-xylosidase (which breaks down xylose), and acid phosphatase (which releases phosphate). Each root tip was placed in an individual well of a 96-well-screen-bottom plate, and enzyme activities were sequentially measured according to the protocol outlined in *Pritsch et al.* (*2011*). Total per-tip activity was calculated based on a standard curve using 4-methylumbelliferone and normalized to surface area calculated using WinRHIZO. Following enzyme assays, DNA was also extracted from these tips, amplified, sequenced, and used either to assign root tips to a fungal taxon or to confirm lack of mycorrhization in the controls.

## Data analysis

We used single-species treatments to verify the viability of our spore inoculum. In multi-species treatments, we determined the competitively dominant fungus as the one with the greatest relative colonization (i.e., greatest proportion of sequences in our randomly selected set of root tips). To determine differences in enzyme activity and seedling growth, we compared treatment means using Tukey's Honestly Significant Difference tests to correct for multiple hypothesis testing. All statistical calculations were performed using R (*R Core Team*, *2014*). To quantify variance in enzymatic activity, we used a principal components analysis to compress data from all five enzymatic assays into two dimensions for visualization (package *bpca*; *Faria, Demetrio & Allaman*, *2016*). We then computed the Euclidean distance between pairs of tips from the same treatments (*vegan*, function *betadisper*; *Oksanen et al.*, *2013*) and performed a Tukey's Honestly Significant Difference test to compare within-treatment variance.

**Table 1  Principal components analysis of enzymatic activity for single-species treatments.**

| Principal component axis | 1 | 2 | 3 | 4 | 5 |
|---|---|---|---|---|---|
| Percent explained | 79.84 | 14.18 | 4.06 | 1.11 | 0.800 |
| Loadings | | | | | |
| $\alpha$-glucosidase | −6.61 | 1.53 | −2.10 | 0.0609 | 0.775 |
| $\beta$-glucosidase | −6.36 | 2.60 | 1.67 | −0.981 | 0.113 |
| N-acetylglucosaminidase | −6.82 | 1.33 | 0.965 | 1.29 | −0.289 |
| $\beta$-xylosidase | −6.77 | −1.60 | −1.19 | −0.437 | −1.01 |
| Acid phosphatase | −5.21 | −4.77 | 0.91 | −0.00634 | 0.568 |

# RESULTS

## Fungal mycorrhization levels
In single-species treatments, all fungi established associations with *P. muricata* seedlings. Mycorrhization levels differed by fungal taxon (Fig. 1). *Thelephora terrestris* had the highest mycorrhization level of 68.2 ± 0.00159% (mean ± standard deviation) root length colonized. *Rhizopogon occidentalis* (23.5 ±0.0952%) and *Suillus pungens* (27.6 ± 15.6%) were similar to one another in abundance. No ectomycorrhizal fungi were detected on the control seedlings.

## Hypothesis 1: dominance hierarchy
Our data supported our hypothesized dominance hierarchy: *T. terrestris* competively excluded both *R. occidentalis* and *S. pungens*. *R. occidentalis* was competitively dominant (in terms of root mycorrhization) to, but did not completely exclude, *S. pungens* in the two-species treatment containing these fungi (Fig. 1).

## Hypothesis 2: dominance-function tradeoff
Enzyme profiles from single-species treatments supported our hypothesis that the least competitive fungus, *S. pungens*, would have the highest enzyme activity levels. *S. pungens* enzymatic activity was higher than non-mycorrhizal control root tips for all five extracellular enzymes assayed (Fig. 2, yellow bars). In contrast, *T. terrestris*, the competitively dominant fungus, had enzymatic activity levels indistinguishable from controls (Fig. 2, blue bars), and *R. occidentalis*, of intermediate competitive ability, had elevated enzymatic activity levels only for $\alpha$- and $\beta$-glucosidase and N-acetyl-glucosaminidase (Fig. 2, red bars). Fungi varied significantly in their overall enzymatic profiles (Fig. 2F, Table 1; see Fig. S1 for additional PCA axes). Variance in enzymatic activity was greatest for *S. pungens*, intermediate for *R. occidentalis*, and lowest (equivalent to control tips) for *T. terrestris* (Fig. 3). In multi-species treatments, $\beta$-Xylosidase and Acid Phosphatase expression levels were elevated for *T. terrestris*-infected tips (Fig. 4).

## Hypothesis 3: seedling growth response
Seedling aboveground, belowground, and total biomass did not differ by treatment; thus our third hypothesis was not supported. Overall, we did observe a positive relationship between mycorrhization level and seedling biomass (Fig. 5).

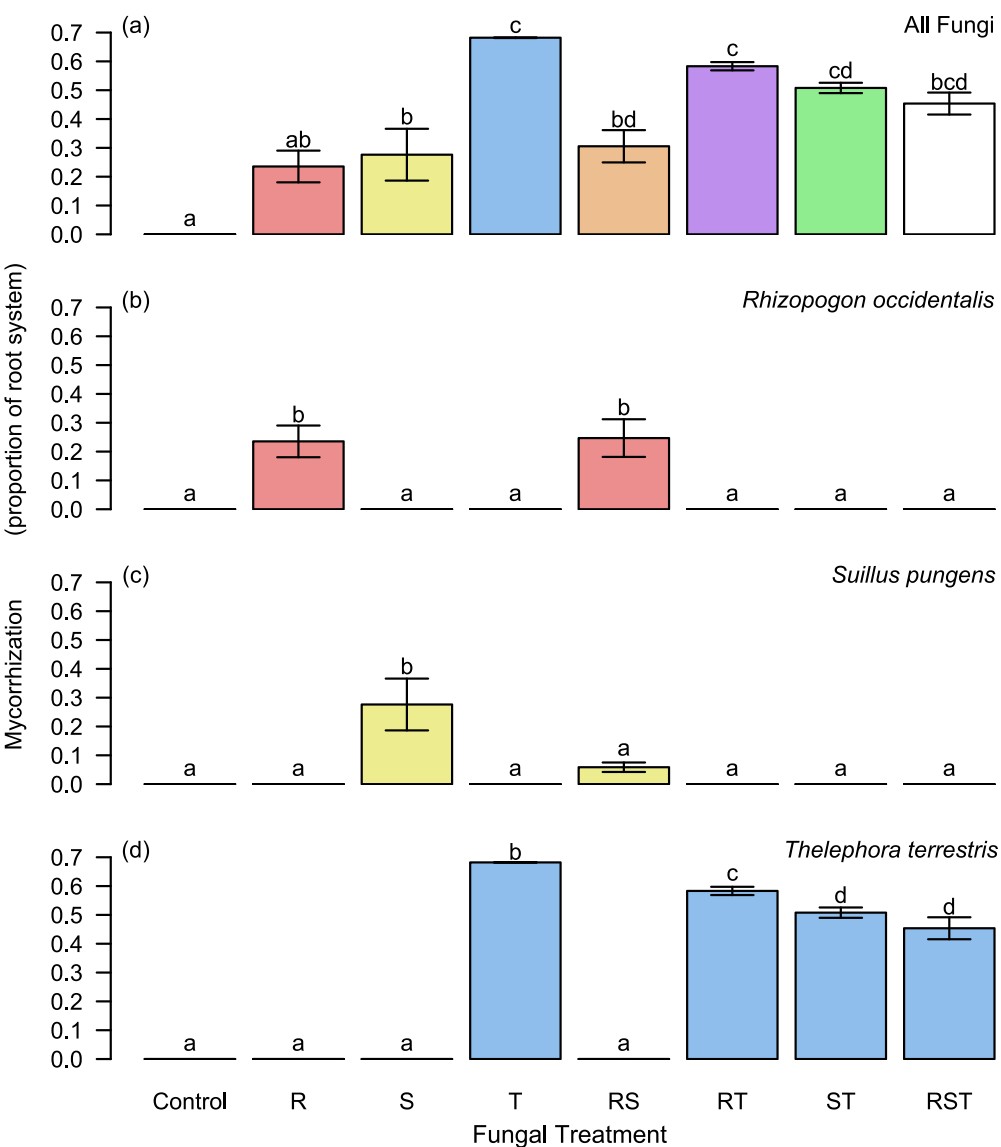

**Figure 1** **Mycorrhization levels across experimental treatments.** (A) Total mycorrhization was highest in treatments that included *Thelephora terrestris*. No fungal contamination was present in the controls. (B–C) Fungi exhibited a dominance hierarchy, with *T. terrestris* as the only fungus present in multi-species treatments, and *Rhizopogon occidentalis* suppressing *Suillus pungens* growth in the two-species combination treatment. Bar heights indicate means across seedlings within a treatment group; whiskers give standard error. Letters indicate statistically significant differences in mean (Tukey's HSD, $P < 0.05$). Colors represent species: Red, *R. occidentalis*, yellow, *S. pungens*, and blue, *T. terrestris*; color blends represent species combinations (e.g., green, *S. pungens* + *T. terrestris* in (A), where total mycorrhization is plotted).

## DISCUSSION

In this study, we present an experimental test for competition-function tradeoffs across three genera of ectomycorrhizal fungi (EMF). We observed a clear competitive dominance hierarchy among the EMF that was inversely related to their extracellular enzymatic activities. Specifically, the most competitively dominant EMF, *Thelephora terrestris*, had

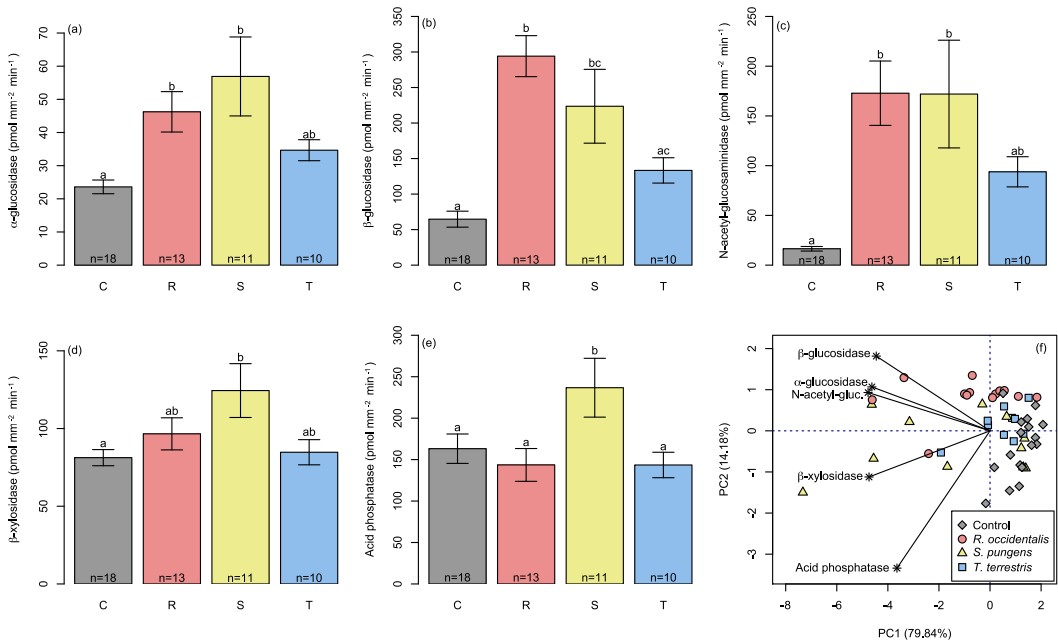

**Figure 2** **Enzyme activities of root tips colonized by the three fungal taxa compared with control (non-mycorrhizal) root tips.** Data are from single-fungus inoculations (i.e., Treatment = R, S, or T). Across the five enzymes tested, only *Suillus pungens*-associated root tips showed consistently elevated activity relative to nonmycorrhizal tips (A–E Tukey's HSD, $P < 0.05$). Association with *Rhizopogon occidentalis* elevated $\alpha$- and $\beta$-glucosidase and N-acetyl-glucosaminidase enzyme activities relative to control tips. A principal component analysis (F) revealed that fungi differed in their enzymatic assays (PERMANOVA, $P < 0.05$).

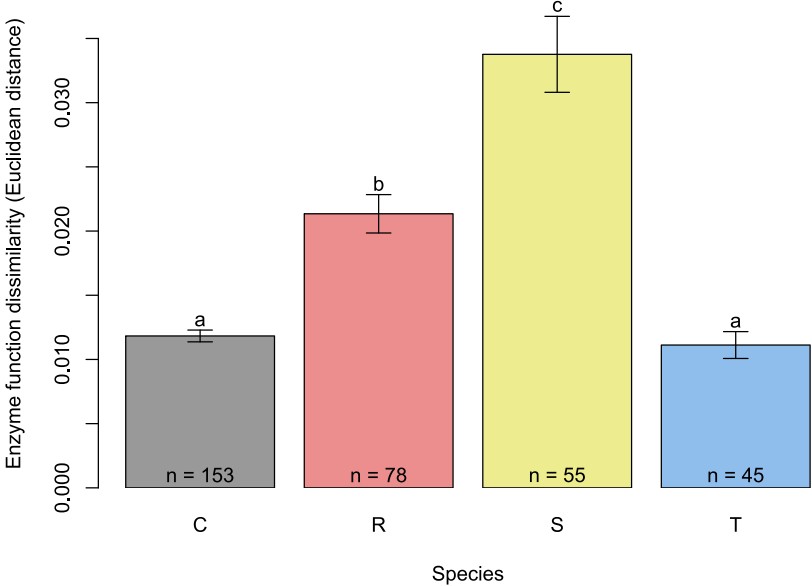

**Figure 3** **Variance in fungal enzymatic activity by species measured as Euclidean distance across all five enzymatic assays.** Data are from single-fungus inoculations (as in Fig. 2). *S. pungens* had the greatest variation in enzymatic function, *R. occidentalis* intermediate, and *T. terrestris* the least (Tukey's HSD, $P < 0.05$).
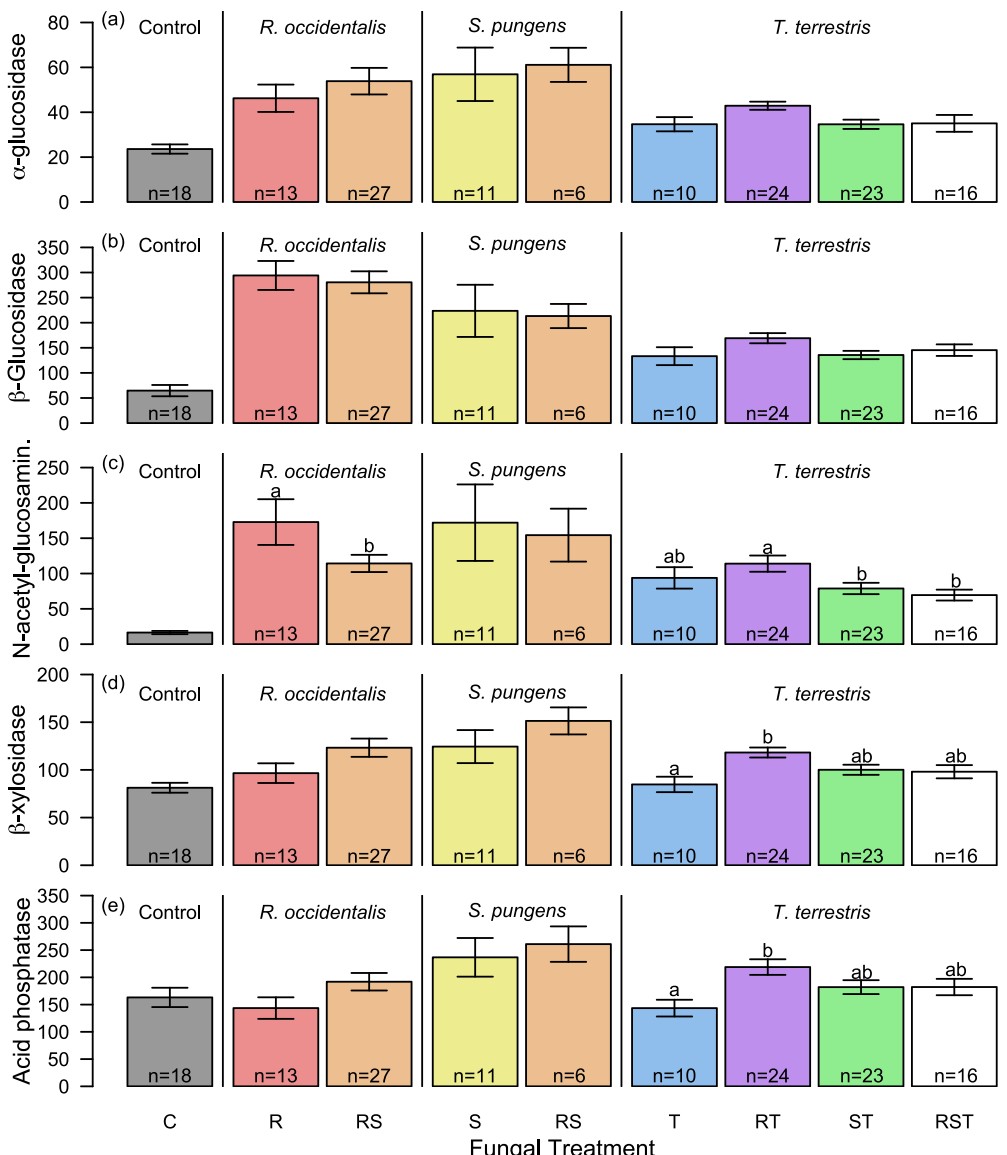

**Figure 4** **Changes in enzyme activity by fungal associate and treatment.** N-acetyl-glucosaminidase activity was elevated for *Rhizopogon occidentalis*-associated root tips when in isolation relative to competition with *Suillus pungens* (C). $\beta$-xylosidase and acid phosphatase activities were elevated for *Thelephora terrestris*-associated tips when in competition with *R. occidentalis* (D–E). Bar heights indicate means across root tips within a treatment group (measured in pmol mm$^{-2}$ min$^{-1}$); whiskers give standard error. Letters indicate statistically significant differences in mean within species by treatment (Tukey's HSD, $P < 0.05$). Colors represent treatments: red, *R. occidentalis*; yellow, *S. pungens*; and blue, *T. terrestris*; color blends represent species combinations (e.g., green, *S. pungens* + *T. terrestris*).

enzymatic activities indistinguishable from non-mycorrhized tree roots, whereas the competitively inferior *Suillus pungens* exhibited elevated enzymatic activity across all five enzymes assayed. A number of studies have previously documented differences in enzymatic activity profiles across (*Courty et al., 2005*; *Courty et al., 2006*; *Buée et al., 2007*; *Courty, Franc & Garbaye, 2010*; *Kipfer et al., 2012*; *Velmala et al., 2013*; *Walker, Ward &*
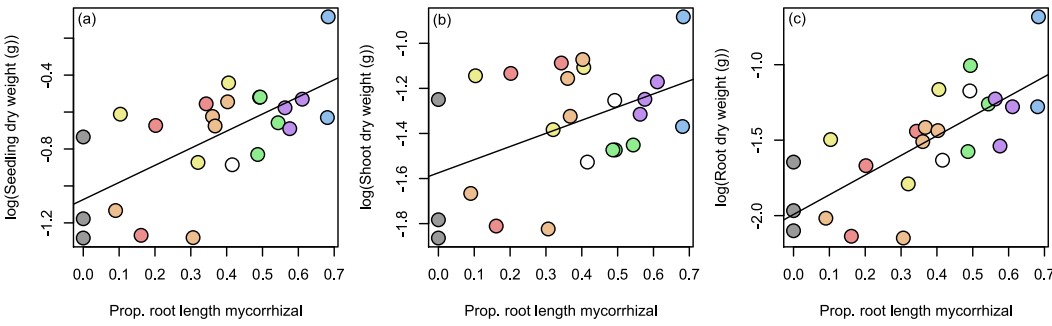

**Figure 5 Relationship between seedling growth and mycorrhization level.** Overall, total seedling biomass was positively correlated with mycorrhization, measured as the proportion of root length with hyphal structures (A, $R^2 = 0.433$, $P < 0.001$). This was the result of both above- and belowground effects of mycorrhization on plant growth: both shoot (B, $R^2 = 0.213$, $P < 0.05$) and root (C, $R^2 = 0.558$, $P < 0.001$) dry weight increased with increasing mycorrhization. Points are color-coded by treatment as in Figs. 1–3.

*Jones*, *2016*) and within (*Jones et al.*, *2010*) fungal genera. These differences, especially when observed in the field among members of assembled EMF communities, are suggestive of functional complementarity (*Courty et al.*, *2005*; *Buée et al.*, *2007*; *Jones et al.*, *2010*). In some cases, e.g., *Rhizopogon* species, enzymatic activities are disproportionately high relative to species abundance (*Walker et al.*, *2014*). However, to our knowledge, this study is the first to experimentally link these differences in enzymatic activity with competitive ability. Because our study, like others that quantify species-specific enzymatic activity, used root tips to obtain fungal tissues, our enzymatic data are representative only of root-tip associated exoenzyme activities, which may be different than expression levels in other parts of the soil (e.g., at the edges of hyphal extent furthest from the host tree where fungi are foraging for nutrients, or at local hotspots of resource availability in the heterogeneous soil environment) (*Wright et al.*, *2005*; *Liao et al.*, *2014*). Thus, much work remains to be done to determine the whole organism's functional potential.

That *T. terrestris* was the most competitively dominant EMF in our study was not surprising given its proclivity for vigorous growth under greenhouse conditions (*Mason et al.*, *1983*; *Velmala et al.*, *2013*). However, *S. pungens* and *R. occidentalis* are key community members on *Pinus muricata* in the field, including at Point Reyes National Seashore where the seeds, soils, and fungal spores used in this study were collected (*Peay et al.*, *2007*). All three species are capable of colonizing seedlings through spore dispersal, with *S. pungens* and *R. occidentalis* more consistently observed on small tree islands than *T. terrestris* (*Peay et al.*, *2007*). *R. occidentalis* colonizes through generation of a long-lived spore bank (*Bruns et al.*, *2009*), while *S. pungens* and *T. terrestris* are the two most prolific aerial spore dispersers in the system (*Peay et al.*, *2012*). Our competitive dominance hierarchy, which is the inverse of this dispersal hierarchy, suggests a competition-colonization tradeoff among these taxa similar to that observed by *Kennedy et al.* (*2011*), at least in terms of root system abundance. While EMF may compete in other ways, such as by competing for nutrient and water resources in the soil or by competing for plant carbon resources (delivery of which may vary by root tip occupancy), in this case the observed complete

exclusion of other EMF by *T. terrestris* suggests strong competitive dominance. However, in cases where exclusion is incomplete measurements of carbon and nitrogen acquisition may be necessary to determine competitive dominance. Although competitively excluded from seedling root systems by *T. terrestris* in our study, the persistence of *S. pungens* and *R. occidentalis* in the field (at least through the first decade of succession) could be the result of several mechanisms. First, less competitive fungi may be maintained through active partner maintenance. This could be the case if trees allocate carbon to their fungal partners in proportion to their partners' provision of resources (*Hoeksema & Kummel*, *2003*), which would likely be greater for highly enzymatically active EMF like *S. pungens*. Second, trees with larger root systems are likely to support a greater diversity of EMF than the seedlings in our study (*Nara et al.*, *2003*). Third, competition among EMF can be context dependent (*Kennedy, Peay & Bruns*, *2009*), and while *T. terrestris* is dominant under greenhouse conditions, this is unlikely to be the case for all biotic and abiotic conditions.

Greenhouse growth conditions may also be responsible for the homogeneity of the seedling growth response. Although *Kipfer et al.* (*2012*) found that the most enzymatically active fungus in their study, *Suillus granulatus*, had a positive effect on seedling growth, we did not observe statistically significant differences in growth by treatment in our study. In part, this was likely due to the homogeneous, high-quality soil environment created by autoclaving the experimental soils, which can release substantial amounts of nutrients into plant-accessible pools, reducing the impact of EMF on seedling growth (*Peay, Bruns & Garbelotto*, *2010*) and plant carbon allocation to EMF (*Hobbie*, *2006*). Other studies have found no relationship between EMF competitiveness and seedling growth (*Kennedy, Peay & Bruns*, *2009*). Perhaps, at least at the early seedling stage, growth effects are obscured by conflicting mechanisms. "High-quality" partners like *S. pungens* may be more expensive thanks to the energetic demands of producing extracellular enzymes; this additional carbon cost may negate any benefits to the host seedling, particularly in nutrient-rich soils.

Although *T. terrestris* was the only fungus observed by harvest time in all treatments that included it as a source of inoculum, we did observe reductions in its mycorrhization level and increases in some of its enzymatic activities in multi-species treatments relative to monoculture. These differences may be due to a time lag in competitive displacement of *R. occidentalis* similar to that observed by *Lilleskov & Bruns* (*2003*) when *R. occidentalis* was in competition with *Tomentella sublilacina* (like *T. terrestris*, a member of the Thelephoraceae), and/or shifts in *T. terrestris* enzymatic function induced by the presence of other EMF. Further experimental manipulation of community composition is likely to elucidate the roles of such mechanisms and clarify the role of fungal identity in mutualism function.

We also observed differences in absolute mycorrhization levels among the EMF in our single-species treatments. By our metric (percent of total root length colonized), *T. terrestris* had mycorrhization levels that were almost double those of the other EMF. In part, this may be due to different root growth forms associated with *R. occidentalis* and *S. pungens*, which tend to induce the production of tightly bunched clusters of root tips (HV Moeller & KG Peay, pers. obs., 2011; see also images in the DEtermination of EctoMYcorrhizae database, http://www.deemy.de/) whose prevalence would be underestimated by our grid-intersect

sampling method. In contrast, *T. terrestris* exhibits greater spatial extent along the length of fine roots, and the formation of root tip clusters is not observed. While this difference in mycorrhization may also be due to interspecific differences in spore inoculum viability or rates of vegetative spread across root systems, prior studies have observed similar levels of mycorrhization for the three genera we studied (*Browning & Whitney*, *1993*; *Kennedy & Bruns*, *2005*). Indeed, *Kennedy & Bruns* (*2005*) observed ∼30% mycorrhization levels for *Rhizopogon* species within two months of inoculation. The evolutionary and ecological reasons for these differences in colonization strategy remain unclear, but a fuller understanding of EMF spatial extent and enzymatic function beyond the plant's immediate root zone will likely shed light on these questions.

## ACKNOWLEDGEMENTS

The authors thank Jennifer Talbot for assistance in performing the enzymatic assays. We also thank Roger Koide and Peter Avis for thoughtful review of our manuscript.

### Funding

HVM received funding from the United States National Science Foundation through a Graduate Research Fellowship, a Doctoral Dissertation Improvement Grant, and a Postdoctoral Research Fellowship in Biology (DBI-1401332). KGP received funding from the National Science Foundation Dimensions of Biodiversity Program (DBI-1045658). The funders had no role in study design, data collection and analysis, decision to publish, or preparation of the manuscript.

### Grant Disclosures

The following grant information was disclosed by the authors:
United States National Science Foundation through a Graduate Research Fellowship.
Doctoral Dissertation Improvement.
Postdoctoral Research Fellowship in Biology: DBI-1401332.
National Science Foundation Dimensions of Biodiversity Program: DBI-1045658.

### Competing Interests

The authors declare there are no competing interests.

### Author Contributions

- Holly V. Moeller and Kabir G. Peay conceived and designed the experiments, performed the experiments, analyzed the data, contributed reagents/materials/analysis tools, wrote the paper, prepared figures and/or tables, reviewed drafts of the paper.

### Data Availability

The raw data has been supplied as Supplemental Information.

## Supplemental Information

Supplemental information for this article can be found online at http://dx.doi.org/10.7717/peerj.2270#supplemental-information.

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
