# Peer review of "Competition-function tradeoffs in ectomycorrhizal fungi"

_PeerJ, doi:10.7717/peerj.2270_

## Round 0.1 · original submission · Minor Revisions

Thank you for a thoughtful contribution. Please consider the suggestions made by these two reviewers. I think that the comments of reviewer 2 wrt spore viability, in particular, merit either some additional data (if available), or else some discussion of how viability may influence assessments of competitiveness.

Please also consider adding some text to figure 1 to explain the color scheme. Although I eventually figured it out, interpreting the color blends was not straight forward.

·

Basic reporting

Excellent

Experimental design

excellent

Validity of the findings

Please see general comments

Additional comments

In this paper the authors test the hypothesis that the competitive ability of three ECM fungi is inversely related to fungal enzymatic activity. Enzymatic function was assessed on a per root tip basis. Competitive ability was taken as the proportion of root tips colonized by the species. The data, as presented, confirm the hypothesis.

The only possible difficulties I see with this interpretation have to do with how competitive ability and enzymatic function were assessed.

Competitive ability was assessed as the proportion of roots tips colonized. This is certainly one measure of competitive ability, but it is not the only. If the acquisition of carbon is the goal of the fungus, then proportion of root tips colonized is a good measure only if carbon transfer rate per root tip is the same for all species. Moreover, if acquisition of water or N is the goal, then colonization of the soil or organic matter in the soil might be a more appropriate measure.

Do we know if enzymatic activity of colonized roots tips is actually related to the ability of the fungus to hydrolyze or oxidize substrates in the soil? The “business end” of the fungus might really be the fine, highly branched hyphae growing within the substrate, not the mantle of hyphae surrounding the root tip. Additionally, enzyme activity is often inducible when substrate is available. If root tips do not have substrate available to them, is their activity necessarily a good indication of the cost of enzyme production?

Perhaps the authors could discuss the ramifications of these observations.

·

Basic reporting

This article is well written in most part, though see below for a few issues. Relevant and sufficient background is included. The structure appears appropriate, though Results need to be "fattened". Figures are solid and relevant.

Experimental design

The design is solid though please note the concern about spore viability will add to what should/needs to be done.

Validity of the findings

The connection between results and hypotheses are clear and (too) concise.

Additional comments

This paper examines the competition vs enzyme production tradeoff of three ectomycorrhizal fungi that associate with pine seedlings. A set of clear hypothesis are tested and the results appear to support most of these. Overall, the paper is well written and clear, but I have a few concerns that I think can be addressed before the paper should be accepted for publication.

The first main concern is spore viability and should be addressed in some way. Either present some spore viability data or make caveats known that the results could be due this instead. The low mycorrhization of the R only and S only treatments makes me concerned. What happens if the viability of spores of the R and S is significantly lower than T? Should another experiment be done to normalize based on equal spore viability? Or, is spore viability simply another trait? Should be discussed somewhere at a minimum.

The second concern is how (impressively) condensed the results section is. Though the figures are appropriate and full of interesting patterns I would expect to be detailed in the results, the results section, as is, is nearly absent (especially given my further comment below). Please more elaboration on these interesting data. For instance, you might consider discussing some of the the variation within species shown in the PCA (e.g. did species cluster along the other axes? As presented, there appear to be some individual reps that drive the vectors). There is plenty to note that is not.

Line comments:

line 89: Why these three taxa? Would be good to know more about when these three taxa are normally found in the same habitat at the same time, in what kinds of numbers? Similar to the outcomes here? Describe more about these three species and the community context in which they exist.

line 121: Were there any spore viability tests done? Although standardized by spore number, should we worry about % germination of spore also? Or should this be considered just another trait. Using mycorrhization in single species treatments does not necessarily do this. A spore viability stain test is preferable.

line 122: Important to explain why this amount of grow time was selected. If longer/shorter growth periods influenced outcome? Would we expect over even longer periods that the more “beneficial” but less “competitive” would increase in numbers?

line 164 (and elsewhere in results): This statement (of support) should be reserved for the discussion section. I prefer to see simple descriptions of data in the results. Perhaps a stylistic issue only, but I think it is a cleaner approach if these statements are left for discussion as the interpretation of support or refute is matter of discussion.

line 172: I suggest modifying “competitively dominant” with “in terms of root mycorrhization”. There could be other means of competition not addressed by this study.

---

## Round 0.2 · accepted · Accept

Thank you for your thoughtful edits. I look forward to seeing it in print.